# Early Oxygen Treatment Measurements Can Predict COVID-19 Mortality: A Preliminary Study

**DOI:** 10.3390/healthcare10061146

**Published:** 2022-06-20

**Authors:** Yosi Levi, Dan Yamin, Tomer Brandes, Erez Shmueli, Tal Patalon, Asaf Peretz, Sivan Gazit, Barak Nahir

**Affiliations:** 1Department of Industrial Engineering, Tel Aviv University, Tel Aviv 69978, Israel; yosilevi@tauex.tau.ac.il (Y.L.); brandes.tomer@gmail.com (T.B.); shmueli@tauex.tau.ac.il (E.S.); 2Center for Combatting Pandemics, Tel Aviv University, Tel Aviv 69978, Israel; 3Kahn Sagol Maccabi (KSM) Research & Innovation Center, Maccabi Healthcare Services, Tel Aviv 68125, Israel; patalon_t@mac.org.il (T.P.); gazit_s@mac.org.il (S.G.); nahirb@gmail.com (B.N.); 4Internal Medicine COVID-19 Ward, Samson Assuta Ashdod University Hospital, Ashdod 7747629, Israel; asafpe@assuta.co.il

**Keywords:** COVID-19 hospitalization, COVID-19 mortality, risk score, inflammatory markers, oxygen

## Abstract

Halting the rapid clinical deterioration, marked by arterial hypoxemia, is among the greatest challenges clinicians face when treating COVID-19 patients in hospitals. While it is clear that oxygen measures and treatment procedures describe a patient’s clinical condition at a given time point, the potential predictive strength of the duration and extent of oxygen supplementation methods over the entire course of hospitalization for a patient death from COVID-19 has yet to be assessed. In this study, we aim to develop a prediction model for COVID-19 mortality in hospitals by utilizing data on oxygen supplementation modalities of patients. We analyzed the data of 545 patients hospitalized with COVID-19 complications admitted to Assuta Ashdod Medical Center, Israel, between 7 March 2020, and 16 March 2021. By solely analyzing the daily data on oxygen supplementation modalities in 182 random patients, we could identify that 75% (9 out of 12) of individuals supported by reservoir oxygen masks during the first two days died 3–30 days following hospital admission. By contrast, the mortality rate was 4% (4 out of 98) among those who did not require any oxygenation supplementation. Then, we combined this data with daily blood test results and clinical information of 545 patients to predict COVID-19 mortality. Our Random Forest model yielded an area under the receiver operating characteristic curve (AUC) score on the test set of 82.5%, 81.3%, and 83.0% at admission, two days post-admission, and seven days post-admission, respectively. Overall, our results could essentially assist clinical decision-making and optimized treatment and management for COVID-19 hospitalized patients with an elevated risk of mortality.

## 1. Introduction

Caused by the severe acute respiratory syndrome coronavirus 2 (SARS-CoV-2), COVID-19 was declared a global pandemic by the World Health Organization (WHO) on 11 March 2020 [1]. It has since challenged healthcare systems worldwide and overwhelmed hospitals and health workers [2], rendering them unable to halt the increased mortality of patients who rapidly deteriorated to perilous health states [3,4]. In hospitals around the globe, the number of patients requiring hospitalization far exceeded the maximum capacity during the pandemic peaks of 2020 and 2021. This reality led to an insufficient number of intensive care unit beds and inadequate treatment, forcing doctors to make difficult ethical decisions [5,6,7].

The rapid clinical deterioration, marked by arterial hypoxemia [8,9], is among the greatest challenges clinicians currently face when treating COVID-19 patients. Severe cases typically feature respiratory distress and require supplemental oxygen and close monitoring. Specifically, supplementation of oxygen therapy for COVID-19 patients is recommended when their peripheral oxygen saturation (SpO2) falls below 94% [10,11]. Moderate-to-severe COVID-19 patients can receive supplemental oxygen, delivered via nose prongs or face masks, through which oxygen flows at a moderate and consistent rate [12]. In contrast, when it comes to patients with acute hypoxemic respiratory failure, a high-flow nasal cannula (HFNC) is preferred [13,14]. Intriguingly, while it is clear that oxygen measures and therapy treatment describe a patient’s clinical conditions at a given time point [15], the potential predictive strength of the duration and extent of oxygen supplementation methods over the entire course of hospitalizations for patient death from COVID-19 has yet to be assessed.

During hospitalization, patients are subjected to various medical tests to monitor disease progression and potential deterioration, including laboratory biomarkers [16]. Malik et al. (2020) identified several such markers of poor outcome, for example, decreased lymphocyte count and elevated levels of C reactive protein, D-dimer, lactate dehydrogenase, and Creatinine kinase [17]. Likewise, Sheth et al. (2021) found distinct Troponin, B-type natriuretic peptide, D-dimer, and Creatinine kinase levels in COVID-19 patients who died or were critically ill compared to less severe disease manifestations [18]. Despite such efforts, no single prognostic biomarker that can distinguish patients who require immediate medical attention and estimate their mortality risk has been identified.

Several pioneering studies have offered machine-learning models to identify respiratory deterioration among hospitalized COVID-19 patients. For example, Gao et al. (2020) developed a COVID-19 machine learning model that predicts the chances of subsequent physiological deterioration and death, based on a patient’s clinical data upon admission [19]. In a similar study, Lassau et al. (2020) constructed a multimodal artificial intelligence severity score that integrates five clinical and biological variables with a deep learning CT-scan model15. Other studies attempted to identify respiratory deterioration based on various biomarkers [20,21,22]. However, these methods did not account for the overall oxygenation levels and the information gained from the physician’s treatment decisions throughout the hospitalization period. Therefore, identifying high-risk patients using practical risk-prediction tools that simultaneously combine all this information should improve the allocation of resources and medical attention.

This study aims to develop a prediction model for COVID-19 mortality in hospitals. We show that integrating clinical, biological, and respiratory data based on oxygen supplementation modalities can be utilized to predict COVID-19 mortality at hospital admission, two days post-admission, and seven days post-admission. Overall, our results may help alert clinicians early on to the elevated mortality risk of a patient during hospitalization and subsequently provide instant intervention and intensive care and monitoring.

## 2. Materials and Methods

### 2.1. Study Population

We retrospectively analyzed the data of 545 patients hospitalized with COVID-19 and admitted to Assuta Medical Center, Ashdod, Israel, between 7 March 2020, and 16 March 2021. For each patient, the data included general information on age, gender, and background diseases and the daily blood test biomarker levels for various inflammation-associated parameters. Each patient in our study received a random ID. Assuta physicians ordered these patients according to their random ID and extracted the respiratory data of the first 182 patients. The extraction process of the oxygen data was performed manually and was time-consuming. Therefore, the medical team could not extract the respiratory data for all 545 patients for our analysis at that time.

### 2.2. Measures

Our data includes the medical information collected from hospital admission to discharge. This includes the daily measurements of oxygen saturation levels and the oxygen supplementation modality, as well as all daily blood test results. All patients were positive for COVID-19 with RT-PCR at admission, but the day of symptom onset or first positive test result was not available. Clinical and laboratory blood test data were obtained from detailed medical records and reviewed, summarized, and cross-checked retrospectively by a team of experienced medical doctors from Assuta Medical Center. Clinical data was comprised of demographic variables (age and gender), and medical history, which included the presence or absence of comorbidities (anemia, COPD, dementia, diabetes, or other). Laboratory blood test data included the neutrophil percentage (NEU%), lymphocytes counts (LYM abs.), total C-reactive protein, D-dimer, LDH in the blood (LDH-B), and Urea in the blood (Urea-B). Respiratory data included (1) oxygen saturation levels (scale of 0–100%); (2) oxygen delivery modalities, ordered by flow rate, including nasal cannula, simple face mask with reservoir, HFNC, Bilevel positive airway pressure (BiPAP) ventilator, and Continuous positive airway pressure (CPAP) ventilator; and (3) flow rates, in liters per minute (LPM).

### 2.3. Oxygenation Score to Predict Severity and Mortality

We generated a single scoring system based on the oxygen supplementation modalities to reflect the severity of the respiratory failure (Table 1), which is consistent with the WHO treatment recommendations [10,11] . This score, which we termed Oxygenation Severity Score (OSS), was scaled from 1 to 10 and was measured daily for each patient. For example, patients who did not receive any form of oxygenation support received a score of 1, whereas patients who required a reservoir oxygen mask with an oxygen flow rate of 1–15 L/min scored 4, and patients who required BiPAP scored 10 (Table 1).

### 2.4. Statistical Analysis

Statistical analysis was performed using Python (version 3.8.5). For descriptive analysis, the median (IQR) was assessed for continuous variables. For our predictive model, we used the Scikit-learn machine learning library.

We developed machine-learning models to predict COVID-19 mortality, utilizing data from three-time points (1) at admission, (2) two days post-admission, and (3) seven days post-admission. In this study, we define “COVID-19 mortality” as a death resulting from COVID-19 associated complications within 30 days post-admission.

For each time point, we analyzed only the data of individuals who were alive at the time of prediction. For each time point, we ran two types of models (1) Naïve–which used only data on age, gender, and the existence of background diseases; (2) Full–which utilized the naïve model data, the daily OSS, and a subset of blood test biomarkers previously shown to be predictive of COVID-19 mortality (see Appendix A). Missing values on respiratory data were imputed by the average of the daily OSS of the individuals who were alive at each time point.

The machine-learning models were evaluated using a Random Forest [20,23] with 500 trees, a maximum depth of 3, a minimal sample of leaves for the splitting of 5, and a minimum impurity decrease of 0.002, based on Gini impurity. Evaluation of the model was conducted using a 100-fold repetition process, where the model was trained each time using 70% of the data and tested over the remaining 30%, in a stratified manner. The reported results are the median of these 100 executions, with a 25 and 75 percentile range. AUC was used as the primary metric to assess the trained models’ overall performance.

To study the contribution of each data source (i.e., naïve data, OSS, and blood test biomarkers), we calculated the incremental contribution to the AUC score by evaluating first the contribution of OSS on top of the naïve model, and then the incremental contribution of the blood test data on top of the OSS model. We chose this order as, typically, information on age, gender, and background disease is available at hospital admission, and OSS is more accessible than blood test biomarkers over the hospitalization period.

## 3. Results

In this study, we analyzed retrospective data of 545 patients who tested positive for COVID-19 and were admitted to the Assuta Medical Center in Ashdod, Israel, due to disease complications, between 7 March 2020, and 16 March 2021. For all those patients the clinical data (age, gender, background diseases) and blood test data that were collected during the hospitalization period, were available. For a random subset of 182 patients, we also extracted respiratory data. These 182 patients had similar, age, gender, and background disease characteristics, as patients for whom respiratory data was not available, as well as similar survival and mortality rates (Table 2). The respiratory data included daily measurements of oxygen saturation levels and the oxygen supplementation modality from the hospitalization records. We examined the data of admitted patients for up to 30 days, while some patients were released before that time, and some died due to COVID-19 complications.

Of the 545 individuals, 58.2% (n = 317) were men. The patients’ ages ranged from 12 to 101 years, with a median age of 67 years (IQR: 54–79). Overall, 16.2% (n = 89) of the hospitalized patients died within 30 days of admission. We found that older age and chronic diseases were associated with elevated mortality risk (Table 2). Specifically, individuals above 80 years of age who were admitted to the hospital due to COVID-19 were 17.2 (95% CI: 6.34–46.43) times more likely to die from disease complications than individuals below the age of 60 (Table 2). Likewise, COVID-19 patients with anemia, chronic obstructive pulmonary disease (COPD), or dementia were 2.6 (95% CI: 1.27–5.35), 3.0 (95% CI: 1.36–6.63), and 3.5 (95% CI: 1.96–6.17) times, respectively, more likely to die than those without these medical conditions.

### 3.1. Oxygenation Score as a Predictor of Survivorship

We aimed to explore the association between respiratory support therapy and mortality of hospitalized COVID-19 patients. We focused our examination of the patient’s data on three time periods (1) at admission, (2) two days post-admission, and (3) seven days post-admission. Regarding the respiratory data, we analyzed the data of a random subset of 182 individuals, for whom respiratory data were available. The extracted respiratory data included daily measurements of oxygen saturation level and the oxygen supplementation modality from the hospitalization records. During the time of hospitalization, the patient could have received one or more of the four available oxygen supplementation modalities, depending on the degree of required aid (from minimal to maximal) (1) nasal canula (2) simple face mask with reservoir (3) HFNC (4) HFNC+reservoir. We found that patients who exhibited more respiratory distress, thereby requiring more supplemental oxygen aid, had less potential to survive the disease. First, we examined the proportion of non-survivors following 2- and 7-days post-admission, depending on the oxygen delivery modality. We revealed a significant difference between patients requiring a face mask with an oxygen reservoir bag attached (herein “reservoir”) and patients who received no oxygen supplementation (herein “no aid”) (Figure 1A,B). Specifically, 75% of individuals supported by reservoir oxygen masks during the first two days died 3–30 days following hospital admission. In stark contrast, for those who did not require any oxygenation supplementation, the mortality rate was significantly less, 4% (Figure 1A). Likewise, 65% of individuals assisted by a reservoir oxygen mask during the first seven days died 8–30 days following hospital admission, compared to 2% of patients who did not require any oxygen supplementation (Figure 1B).

Owing to the predictive potential of the respiratory stress on the survival rate, we developed a simple metric (1–10 scale), which we termed Oxygenation Severity Score (OSS), to determine the clinical severity of a COVID-19 patient (Table 1). This score was determined daily for each patient based on the oxygen supplementation modality/delivery method and the flow rate used. For example, patients who did not receive supplemental oxygen were assigned a score of 1, whereas those who required a reservoir oxygen mask with a 1–15 L/min oxygen flow scored 4, and those who required a BiPAP scored 10. We observed that, while the OSS increased with the length of hospitalization, non-survivors had a substantially higher OSS at an earlier stage of their hospitalization (Figure 1C). Specifically, 38% of patients who reached an OSS ≥4 on the first or second day of hospitalization did not survive (RR: 2.5, CI 95%:1.2–5). Additionally, we found that a higher OSS score correlated with less time until death (in days) (Figure 1D). For example, 78% of diseased patients had a score ≥4 more than 2 days before death, and 55% of diseased patients had a score ≥4 more than 4 days before death. Therefore, the OSS could provide an early sign of elevated risk of mortality.

### 3.2. Integration of OSS with Biological Blood Test Data to Predict Mortality

After analyzing the clinical data (age, gender, and background diseases) and respiratory data, we aimed to create a predictive model for mortality from COVID-19. We also accounted for the laboratory blood test data collected from patients during the hospitalization period. We developed two Random Forest models, one that utilized only information on age, gender, and background diseases, and another that used all the data, including OSS and blood test biomarkers. Our model included all 545 patients. However, only a random subset of 182 patients had available respiratory data. Thus, as the Random Forrest algorithm cannot use missing values, we imputed the missing OSS values for the remaining 363 patients by the average of the daily OSS of the individuals who were alive at the time of prediction. This procedure, which uses the daily average values in patients for whom data is not available, is equivalent to adding a ‘null’ value as it does not add any information on patients with no OSS data. Of note, the 182 patients had similar survival and mortality rates to the remaining 363 patients, for whom the respiratory data was not available. We computed the performance of our models to determine COVID-19-associated mortality following (1) hospital admission, (2) two days post-admission, and (3) seven days post-admission. We compared the results of the three prediction scenarios to a naïve model that relied solely on the information before hospitalization (i.e., gender, age, and background diseases).

We observed that the 2- and 7-day predictive models, which combine the clinical, OSS, and blood test data, better-classified patients into survivors and non-survivors than the naïve model (Figure 2A). Namely, our integrated models achieved an area under the receiver operating characteristic curve (AUC) score on the test set of 82.5%, 81.3%, and 83.0% at admission, two days post-admission, and seven days post-admission, respectively. The longer the hospital stay, the greater the OSS and blood tests contributed to predicting mortality (Figure 2B). Furthermore, we observed that several blood tests correlated with the severity of the disease by diverging significantly between survivors and non-survivors. Examples of such tests are neutrophil percentage (NEU%), lymphocyte count (LYM abs.), blood urea (Urea-B), and blood LDH (LDH-B) levels (Appendix A). Nevertheless, we observed no correlation between the laboratory blood test markers and the OSS (Appendix A), emphasizing why the OSS provided additional predictive information for COVID-19 mortality beyond that extracted from laboratory blood tests. Therefore, combining biological data with the respiratory data could allow clinicians to target high-risk patients during hospitalization to provide instant intervention and prevent mortality due to COVID-19 associated complications.

## 4. Discussion

Rapid clinical deterioration is among the greatest challenges clinicians face when treating COVID-19 patients in hospitals. To allow better early detection of such deterioration, we have developed a simple scoring system based on oxygenation data to predict the mortality of hospitalized COVID-19 patients. We found that combining this simple scoring system with basic information on age and background diseases can predict the potential risk of mortality during hospitalization. Integrating blood tests into our model further improved our predictions, with an AUC score of 82.5%, 81.3%, and 83.0% at admission, two days post-admission, and seven days post-admission, respectively. Overall, the combination of both medical and respiratory parameters could assist clinicians in targeting high-risk patients during hospitalization, assist in their decision-making process, and allow for rapid intervention to prevent mortality caused by COVID-19 associated complications.

COVID-19 disease progression is monitored by laboratory blood tests during the patient hospitalization period [16,17,18,24,25]. The biomarkers used can be divided into several groups: hematological (i.e., neutrophil and lymphocyte count), inflammatory (i.e., C-reactive protein), immunological (i.e., IL-6), and biochemical (D-dimer, LDH, urea). Our examination of biomarkers from the different groups showed a clear difference between survivors and non-survivors, mainly in patients with an elevated neutrophil percentage (NEU%), decreased lymphocyte count (LYM abs.), and lower urea and LDH levels in the blood (urea-B and LDH-B, respectively). Indeed, no single prognostic biomarker could estimate a patient’s mortality risk. Therefore, combining multiple markers enabled a more accurate identification of the patients’ likelihood of developing severe symptoms of respiratory failure. Conversely, blood test information is retrospective to the time of measurement. By contrast, the OSS is a simple measure that can be recorded non-invasively in real-time. Although each layer of information contributes equally to the predictive models, combining continuous OSS measures early during hospitalization and detailed blood laboratory biomarkers can serve as a powerful tool for early detection of patients’ deterioration, allowing for rapid and efficient treatment.

Our study has several limitations that should be addressed. First, our data includes clinical and biological information on 545 patients, but the respiratory data is limited to 182 patients. Our results should be validated with larger samples. Second, the data were retrospectively obtained from a single medical center, with its particular medical practices, which may decrease the ability to generalize the results, especially considering the COVID-19 variability among populations and countries [26]. Third, our data was collected during 2020/1 at the first outbreak of the COVID-19 pandemic. Since then, several variants of the virus have emerged, with different onsets of clinical symptoms and mortality rates. Nevertheless, we denote that the treatment practice used at the hospital has not considerably changed since then and the isolation of high-risk patients for respiratory deterioration remains a challenge in health care systems. Therefore, the ability to identify patients with high mortality risk is essential for clinical decision-making and optimized treatment and management.

Despite the recent development of effective vaccines, the coronavirus pandemic will likely continue to affect our lives in the years to come due to the emergence of new, highly transmissible mutant strains [27,28,29], the incomplete efficiency of the developed vaccines in the elderly [30] and high-risk populations, and the objection to vaccinating certain populations, such as individuals with a history of allergies [31]. Thus, in the absence of medically approved treatments, using practical risk-prediction tools, such as demonstrated in this study, may help ensure the provision of adequate clinical care to COVID-19 patients in a critical state and intensive monitoring during the hospitalization period.

In conclusion, we provide a new dynamic machine-learning model that combines data derived from monitoring medical and respiratory parameters. The model integrates clinical and biological information to alert at hospital admission, two days post-admission, and seven days post-admission against patients with an elevated risk of dying. Overall, our model could essentially assist clinical decision-making and intensive monitoring of COVID-19 infected individuals with unfavorable prognostic indicators and a high risk of mortality from disease complications.

## Figures and Tables

**Figure 1 healthcare-10-01146-f001:**
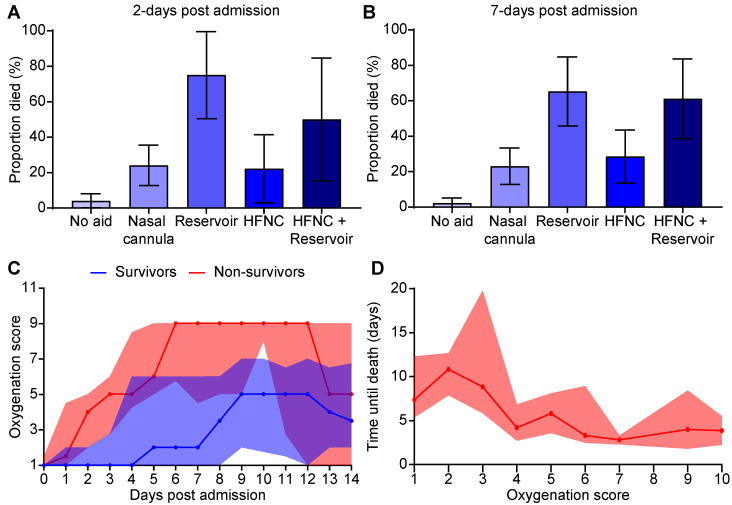
(**A**) The proportion of individuals who died 3–30 days post-admission to the type of oxygenation aid treatment provided during the first two days post-admission. (**B**) The proportion of individuals who died 7–30 days post-admission to the type of oxygenation aid treatment provided during the first seven days post-admission. Error bars represent the 95% confidence interval. (**C**) Daily median oxygenation score of survivors and non-survivors admitted to the hospital. The purple area represents the overlapping between the 95% confidence interval of the “Survivors” and “Non survivors” graphs. (**D**) The number of days until death as a function of the maximal daily Oxygenation Severity Score (OSS). The light red/blue areas represent the interquartile range.

**Figure 2 healthcare-10-01146-f002:**
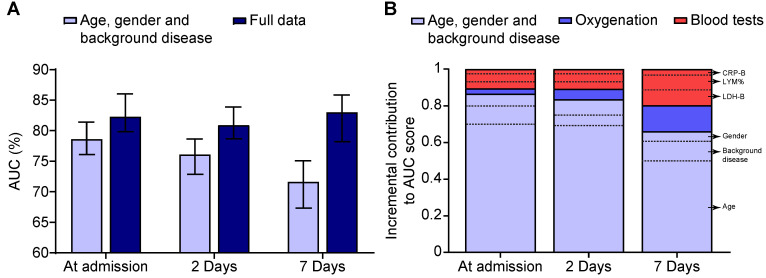
Predictive models’ performance. (**A**) Mean AUC of a model that utilizes data before hospital admission (i.e., age, gender, and background diseases) and of a model that utilizes data before and during hospitalization (i.e., includes the oxygenation score and blood biomarkers). AUC scores are presented for patients on the day of admission, two days post admission, and seven days post admission. (**B**) Sociodemographic and background disease, oxygenation, and blood test data and their sequential contribution to the “at admission”, “2-days” post admission and “7-days” post admission predictive models.

**Table 1 healthcare-10-01146-t001:** Oxygenation Severity Score.

Respiratory Aid	Liters per Minute (LPM)	Oxygenation Severity Score
Room air	0	1
Nasal cannula	1–4	2
Nasal cannula	5–10	3
Reservoir	1–15	4
Reservoir	15–20	5
HFNC	31–40	6
HFNC	20–30	7
HFNC	<20	8
HFNC + Reservoir		9
BiPAP/CPAP		10

**Table 2 healthcare-10-01146-t002:** Information on hospitalized patients with COVID-19 between 7 March 2020, and 16 March 2021, at Assuta Ashdod Medical Center.

Category Type	Category	Respiratory Data	N	Survivors %	Survivors	Non Survivors	Non Survivors %	Relative Risk	95% Confidence Level
Age	12–60	True	69	67	97.1%	2	2.9%		
		False	115	113	98.3%	2	1.7%		
		Both	184	180	97.8%	4	2.2%	–	–
	60–80	True	72	62	86.1%	10	13.9%		
		False	163	138	84.7%	25	15.3%		
		Both	235	200	85.1%	35	14.9%	6.85 ^A*^	2.48–18.93
	>80	True	41	26	63.4%	15	36.6%		
		False	85	53	62.4%	32	37.6%		
		Both	126	79	62.7%	47	37.3%	17.16 ^A*^	6.34–46.43
Gender	Female	True	64	54	84.4%	10	15.6%		
		False	164	141	86.0%	23	14.0%		
		Both	228	195	85.5%	33	14.5%	0.87	0.58–1.29
	Male	True	118	101	85.6%	17	14.4%		
		False	199	163	81.9%	36	18.1%		
		Both	317	264	83.3%	53	16.7%	–	–
Background diseases	None	True	60	54	90.0%	6	10.0%		
		False	112	96	85.7%	16	14.3%		
		Both	172	150	87.2%	22	12.8%	–	–
	Anemia	True	7	4	57.1%	3	42.9%		
		False	14	10	71.4%	4	28.6%		
		Both	21	14	66.7%	7	33.3%	2.61 ^B*^	1.27–5.53
	COPD	True	4	4	100.0%	0	0.0%		
		False	9	4	44.4%	5	55.6%		
		Both	13	8	61.5%	5	38.5%	3.01 ^B*^	1.36–6.63
	Dementia	True	13	6	46.2%	7	53.8%		
		False	14	9	64.3%	5	35.7%		
		Both	27	15	55.6%	12	44.4%	3.47 ^B*^	1.96–6.17
	Diabetes	True	37	28	75.7%	9	24.3%		
		False	83	70	84.3%	13	15.7%		
		Both	120	98	81.7%	22	18.3%	1.43 ^B^	0.83–2.47
	Other	True	73	67	91.8%	6	8.2%		
		False	145	123	84.8%	22	15.2%		
		Both	218	190	87.2%	28	12.8%	1.0 ^B^	0.6–1.69

^A^ Relative risk of the computed group compared to age group 12–60. ^B^ Relative risk of the computed group compared to individuals with no background diseases. * Statistically significant at *p* < 0.05, Chi-square test of independence.

## Data Availability

Access to the data used for this study can be made available upon request and is subject to internal review approval from the institutional review board of MHS with the current data sharing guidelines of MHS and Israeli law.

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
