# Peer review of "Early Oxygen Treatment Measurements Can Predict COVID-19 Mortality: A Preliminary Study"

_healthcare, 2022, doi:10.3390/healthcare10061146_

Round 1

Reviewer 1 Report

Halting the rapid clinical deterioration, marked by arterial hypoxemia, is among the greatest challenges clinicians face when treating COVID-19 patients in hospitals. While it is clear that oxygen measures and treatment procedures describe a patient’s clinical condition at a given time point, the potential predictive strength of the duration and extent of oxygen supplementation methods over the entire course of hospitalization for patient death from COVID-19 has yet to be assessed.

The authros: (a)  introduced a single scoring system based on oxygen supplementation modalities to predict COVID-19 mortality in hospitals. (b) analysed the data on oxygen supplementation methods in 182 patients hospitalized due to COVID-19 complications admitted to Assuta Ashdod Medical Center, Israel, between March 7, 2020, and March 16.

They found that the maximal daily score reported since hospital admission provides an early sign of elevated risk of mortality >48 hours after the alert in nearly 80% of the patients.

They also utilized this data with blood test results of 545 patients in total to predict COVID-19 mortality.

Their  model yielded an area under the receiver operating characteristic curve (AUC) score on the test set of 82.5%, 81.3%, and 83.0% at admission, two days post admission, and seven days post admission, respectively.

Overall, their work can potentially assist clinicians in identifying high-risk patients early on, which is essential for clinical decision-making and optimized treatment and management.

Interesting paper.

This contribute is particularly useful in the pandemic era.

It needs some improvements before being accepted.

Some comments for the authors:

  1. Check the verbs. Starting from the abstract sometimes you use the past tense and other times you use the present tense.
  2. Better structure the text at the end of the introduction”“Here, we introduce a single scoring system based on oxygen supplementation modalities to predict COVID-19 mortality in hospitals. We demonstrate how we can predict COVID-19 mortality at hospital admission, two days post admission, and seven days post admission by integrating clinical (age, gender, background diseases), respiratory and blood tests data into a machine-learning algorithm. Overall, our results may help alert clinicians early on to the elevated mortality risk of a patient during hospitalization.” It is not appropriate for a purpose.
  3. Usually the methods comprehend also a subsection “measures”. Please add it
  4. Results are divided by themes. Add some rows justifying the themes.
  5. The conclusions at the end of the discussion should better highlight the significance of the study and the efforts of the authors. Perhaps you could separate add some text from introduction (see comment 2).
  6. Check tables. They are not following the MDPI standards
  7. Check the figures. Is the resolution suitable?
  8. Minimize the use of the acronyms

Author Response

Dear reviewer,

Thank you very much for your valuable comments for our manuscript entitled “Early oxygen treatment measurements can predict COVID-19 mortality: a preliminary study”.

We enclose our revised manuscript, modified according to your comments. As you will see below, we have addressed all of your points and provided a point-by-point response that includes the comment (in bold) followed by the response (in italic).

We hope our revisions provide complete answers to the questions raised and satisfy the requirements for publication. Thank you very much for your consideration.

Sincerely,

Dan Yamin, PhD

Head, Laboratory for Epidemic Modeling and Analysis

Faculty of Engineering, Tel-Aviv University

Interesting paper. This contribute is particularly useful in the pandemic era. It needs some improvements before being accepted.

Response: We thank the reviewer for finding our study interesting and useful, and we highly appreciate the thoughtful comments.

Some comments for the authors:

1. Check the verbs. Starting from the abstract sometimes you use the past tense and other times you use the present tense.

Response: We revised the entire manuscript text and fixed all verbs and other misspelling mistakes.

2. Better structure the text at the end of the introduction “Here, we introduce a single scoring system based on oxygen supplementation modalities to predict COVID-19 mortality in hospitals. We demonstrate how we can predict COVID-19 mortality at hospital admission, two days post admission, and seven days post admission by integrating clinical (age, gender, background diseases), respiratory and blood tests data into a machine-learning algorithm. Overall, our results may help alert clinicians early on to the elevated mortality risk of a patient during hospitalization.” It is not appropriate for a purpose.

Response: We agree with the reviewer and are thankful for this point. We, therefore, revised the final paragraph in the introduction section and clarified the purpose of the study in the abstract as well. The revised paragraphs are as follows:

Abstract: “In this study, we aim to develop a prediction model for COVID-19 mortality in hospitals by utilizing data on oxygen supplementation modalities of patients.

Introduction: (Page 2, line 64) “In this study, we aim to develop a prediction model for COVID-19 mortality in hospitals. We show that integrating clinical, biological, and respiratory data based on oxygen supplementation modalities can be utilized to predict COVID-19 mortality at hospital admission, two days post-admission, and seven days post-admission. Overall, our results may help alert clinicians early on to the elevated mortality risk of a patient during hospitalization and subsequently provide instant intervention, as well as intensive care and monitoring.”

3. Usually the methods comprehend also a subsection “measures”. Please add it

Response: Following the reviewer’s comment, we now added the “measures” subsection to the Materials and Methods section:

(Page 2, line 81) “Our data includes the medical information collected from hospital admission to discharge. This includes the daily measurements of oxygen saturation levels and the oxygen supplementation modality as well as all daily blood test results. All patients were positive for COVID-19 with RT-PCR at admission, but the day of symptom onset or first positive test result was not available. Clinical and laboratory blood test data were obtained from detailed medical records and reviewed, summarized, and cross-checked retrospectively by a team of experienced medical doctors from Assuta Medical Center. Clinical data was comprised of demographic variables (age and gender), and medical history, which included the presence or absence of comorbidities (anemia, COPD, dementia, diabetes, or other). Laboratory blood test data included the neutrophil percentage (NEU%), lymphocytes counts (LYM abs.), total C-reactive protein, D-dimer, LDH in the blood (LDH-B), and Urea in the blood (Urea-B). Respiratory data included (1) oxygen saturation levels (scale of 0-100%), (2) oxygen delivery modalities, ordered by flow rate, including nasal cannula, simple face mask with reservoir, HFNC, Bilevel positive airway pressure (BiPAP) ventilator, and Continuous positive airway pressure (CPAP) ventilator, and (3) flow rates, in liters per minute (LPM).”

4. Results are divided by themes. Add some rows justifying the themes.

Response: We added the following sentence in the Results section:

(Page 5, line 158) “We aimed to explore the association between respiratory support therapy and mortality of hospitalized COVID-19 patients.”

5. The conclusions at the end of the discussion should better highlight the significance of the study and the efforts of the authors. Perhaps you could separate add some text from introduction (see comment 2).

Response: We agree with the reviewer and are thankful for this point. We, therefore, revised the last paragraph in the Discussion section:

(Page 8, line 278) “In conclusion, we provide a new dynamic machine-learning model that combines data derived from monitoring medical and respiratory parameters. The model integrates clinical and biological information to alert at hospital admission, two days post-admission, and seven days post-admission against patients with elevated risk to die. Overall, our model could essentially assist clinical decision-making and intensive monitoring of COVID-19 infected individuals with unfavorable prognostic indicators, and a high risk of mortality from disease complications.”

6. Check tables. They are not following the MDPI standards

Response: We agree with the reviewer and therefore modified Table 1 and Table 2 to the appropriate MDPI standards.

7. Check the figures. Is the resolution suitable?

Response: We thank the reviewer for his comment. We changed all figures from JPEG format to high-resolution PDF format.

8. Minimize the use of the acronyms

Response: We agree with the reviewer and are thankful for this point. As mentioned in ”point 1”, we revised the entire manuscript text and minimized the use of unnecessary acronyms.

Reviewer 2 Report

Early oxygen treatment measurements can predict COVID-19 mortality: a preliminary study

I thank you for the opportunity to comment this article. The topic is interesting. I have although several concerns related to this preliminary study.

Comments:

-Authors conclusion is the following:

In conclusion, we provide a new dynamic machine-learning model that combines data derived from the monitoring of both medical and respiratory parameters. The model integrates clinical and biological information and can potentially assist clinicians to target high-risk patients during hospitalization by quickly and accurately predicting respiratory deterioration, essential for clinical decision-making and optimized treatment and 248 management.

The conclusion remains quite abstract. Authors need to carefully tell how this can improve decision making and how the treatment could be optimized?

  • I wonder why this material collected in March 2020 is not published earlier?
  • In 2020 the clinical features were different of COVId-19 because of the different mutation types. This issue needs to be discussed and mentioned also in the limitations.
  • I don’t find any description how many days these hospitalized patients did have SARS-COV-2 infection at the time point when the first measurement was carried out. This is very crucial question. Kindly provide this data. Were these measurements carried out during several days or just one measurement?
  • Authors had complete data set only regarding 182 patients. in 3.2 authors say that they imputed missing OSS values for the remaining 363 patients by the average of the daily OSS of the individuals who were live at the time of prediction. I am not satisfied with this methodology and suggest strongly that authors use only the patient data which is available.

Author Response

Dear reviewer,

Thank you very much for your valuable comments for our manuscript entitled “Early oxygen treatment measurements can predict COVID-19 mortality: a preliminary study”.

We enclose our revised manuscript, modified according to your comments. As you will see below, we have addressed all of your points and provided a point-by-point response that includes the comment (in bold) followed by the response (in italic).

We hope our revisions provide complete answers to the questions raised and satisfy the requirements for publication. Thank you very much for your consideration.

Sincerely,

Dan Yamin, PhD

Head, Laboratory for Epidemic Modeling and Analysis

Faculty of Engineering, Tel-Aviv University

I thank you for the opportunity to comment this article. The topic is interesting. I have although several concerns related to this preliminary study.

Response: We thank the reviewer for finding our study interesting, and we highly appreciate the thoughtful comments.

Comments:

1. Authors conclusion is the following:

In conclusion, we provide a new dynamic machine-learning model that combines data derived from the monitoring of both medical and respiratory parameters. The model integrates clinical and biological information and can potentially assist clinicians to target high-risk patients during hospitalization by quickly and accurately predicting respiratory deterioration, essential for clinical decision-making and optimized treatment and management. The conclusion remains quite abstract. Authors need to carefully tell how this can improve decision making and how the treatment could be optimized?

Response: We agree with the reviewer and are thankful for this point. We, therefore, revised the final paragraph in the Discussion section and clarified how our dynamic model can help physicians in decision-making and how their treatment could be optimized. The revised paragraph is as follows:

(Page 8, line 278) “In conclusion, we provide a new dynamic machine-learning model that combines data derived from monitoring medical and respiratory parameters. The model integrates clinical and biological information to alert at hospital admission, two days post-admission, and seven days post-admission against patients with elevated risk to die. Overall, our model could essentially assist clinical decision-making and intensive monitoring of COVID-19 infected individuals with unfavorable prognostic indicators, and a high risk of mortality from disease complications.”

2. I wonder why this material collected in March 2020 is not published earlier? In 2020 the clinical features were different of COVId-19 because of the different mutation types. This issue needs to be discussed and mentioned also in the limitations.

Response: We agree with the reviewer. We, therefore, revised the third paragraph in the Discussion section to now address the limitations of the emergence of new COVID-19 mutants since we first conducted our study. The revised paragraph is as follows:

(Page 8, line 256) “Our study has several limitations that should be addressed. First, our data includes clinical and biological information on 545 patients, but the respiratory data is limited to 182 patients. Our results should be validated with larger samples. Second, the data were retrospectively obtained from a single medical center, with its particular medical practices, which may decrease the ability to generalize the results, especially considering the COVID-19 variability among populations and countries[1]. Third, our data was collected during 2020/1 at the first outbreak of the COVID-19 pandemic. Since then, several variants of the virus have emerged, with different onsets of clinical symptoms and mortality rates. Nevertheless, we denote that the treatment practice used at the hospital has not considerably changed since then and isolation of high-risk patients for respiratory deterioration remains a challenge in health care systems. Therefore, the ability to identify patients with high mortality risk is essential for clinical decision-making and optimized treatment and management.”

3. I don’t find any description how many days these hospitalized patients did have SARS-COV-2 infection at the time point when the first measurement was carried out. This is very crucial question. Kindly provide this data. Were these measurements carried out during several days or just one measurement?

Response: Our data includes the medical information collected from hospital admission to discharge. This includes the daily measurements of oxygen saturation levels and the oxygen supplementation modality as well as all daily blood test results. All patients were positive for COVID-19 with RT-PCR at admission, but the day of symptom onset or first positive test result is not available. For complete transparency, we now added this clarification to the subsection “5.3 meseures” in the Methods.

4. Authors had complete data set only regarding 182 patients. in 3.2 authors say that they imputed missing OSS values for the remaining 363 patients by the average of the daily OSS of the individuals who were live at the time of prediction. I am not satisfied with this methodology and suggest strongly that authors use only the patient data which is available.

Response: We would like to emphasize that we had available respiratory data only for a subset of 182 patients that were randomly selected by a team of experienced medical doctors from Assuta Medical Center. Furthermore, it can be seen in Table 2 that the 182 patients had similar survival and mortality rates to the remaining 363 patients for whom the respiratory data was not available. We added the following sentences in the Results section to emphasize these points:

(Page 4, line 140) “These 182 patients had similar age, gender, and background disease characteristics, as patients for whom respiratory data was not available, as well as similar survival and mortality rates (Table 2)”

(Page 7, line 204) “Of note, the 182 patients had similar survival and mortality rates to the remaining 363 patients, for whom the respiratory data was not available.”

References

  1. Chen L, Zheng S (2020) Understand variability of COVID-19 through population and tissue variations in expression of SARS-CoV-2 host genes. Informatics Med Unlocked 21:. https://doi.org/10.1016/j.imu.2020.100443

Round 2

Reviewer 1 Report

The manuscript improved.

There are not further comments.

Author Response

Dear reviewer,

Thank you very much for your valuable comments for our manuscript entitled “Early oxygen treatment measurements can predict COVID-19 mortality: a preliminary study”.

Reviewer 2 Report

Thank you for the comments.

Author Response

(The authors gave the same response as above.)
